# Acquisition of handwriting in children with and without dysgraphia: A computational approach

Thomas Gargot [1,2,3] *, Thibault Asselborn[4], Hugues Pellerin[1], Ingrid Zammouri[1], Salvatore M. Anzalone[3], Laurence Casteran[5], Wafa Johal[6], Pierre Dillenbourg[4], David Cohen[1,2], Caroline Jolly [7,8]

**1** Psychiatrie de l'Enfant et de l'Adolescent, Pitié Salpêtrière—Charles Foix, Assistance Publique Hôpitaux de Paris, Paris, France, **2** ISIR, Sorbonne Université, Paris, France, **3** CHART Laboratory—EA 4004, TIM, Paris 8 University, Saint Denis, France, **4** CHILI Lab, EPFL University, Route Cantonale, Lausanne, Switzerland, **5** Reference Center for Speech and Learning Disorders, Grenoble Hospital, Grenoble, France, **6** University of New South Wales, Sidney, Australia, **7** LPNC, Univ. Grenoble Alpes, Grenoble, France, **8** CNRS, LPNC UMR 5105, Grenoble, France

* thomas.gargot@etud.univ-paris8.fr

**Data Availability Statement:** The data set that served for statistical analyses performed in the manuscript are available at the following repository (http://osf.io/d845e). The raw data set cannot be

## Abstract

Handwriting is a complex skill to acquire and it requires years of training to be mastered. Children presenting dysgraphia exhibit difficulties automatizing their handwriting. This can bring anxiety and can negatively impact education. 280 children were recruited in schools and specialized clinics to perform the Concise Evaluation Scale for Children's Handwriting (BHK) on digital tablets. Within this dataset, we identified children with dysgraphia. Twelve digital features describing handwriting through different aspects (static, kinematic, pressure and tilt) were extracted and used to create linear models to investigate handwriting acquisition throughout education. K-means clustering was performed to define a new classification of dysgraphia. Linear models show that three features only (two kinematic and one static) showed a significant association to predict change of handwriting quality in control children. Most kinematic and statics features interacted with age. Results suggest that children with dysgraphia do not simply differ from ones without dysgraphia by quantitative differences on the BHK scale but present a different development in terms of static, kinematic, pressure and tilt features. The K-means clustering yielded 3 clusters (Ci). Children in C1 presented mild dysgraphia usually not detected in schools whereas children in C2 and C3 exhibited severe dysgraphia. Notably, C2 contained individuals displaying abnormalities in term of kinematics and pressure whilst C3 regrouped children showing mainly tilt problems. The current results open new opportunities for automatic detection of children with dysgraphia in classroom. We also believe that the training of pressure and tilt may open new therapeutic opportunities through serious games.

shared publicly because of ethical restrictions. Even if the data are de-identified, part of them have been acquired in a medical context and contain sensitive patient information. In addition, the parents of all participants did not give an explicit consent to disclose publicly the basic demographic information and writing tests of their child. The ethics agreement obtained from the University Ethics Committee for this study did not authorize public disclosure of participants' data. We share here an example of writing of a child whose parents gave informed consent for sharing. The complete dataset can be obtained upon request from LPNC: caroline.jolly@univ-grenoblealpes.fr or Psychiatrie de l'Enfant et de l'Adolescent, Pitié Salpêtriére - Charles Foix: hugues.pellerin@aphp.fr and isabelle.babilaere@aphp.fr

**Funding:** We would like to thank the Swiss National Science Foundation for supporting this project through the National Center of Competence in Research Robotics. We would also like to thank the Assistance Publique Hôpitaux de Paris, the Paris 8 University and the Centre National de la Recherche Scientifique (CNRS) for financial support and the ANR Grant (ANR-19-CE19-0029) and the SNSF for support of the project IReCheck.

**Competing interests:** The authors have declared that no competing interests exist.

## Introduction

Handwriting is an essential skill, since children spend up to 60% of their time at school writing [1]. Appropriately legible and automated handwriting is necessary for the acquisition of other higher-order skills such as spelling and story composition. Handwriting is a complex perceptual–motor task, as it involves attention, perceptual, linguistic and fine motor skills [2–7].

Formal handwriting acquisition begins at the age of five years (preschool) and requires about ten years of practice to reach a level of almost complete automation [3, 8–14]. During this time, handwriting initially evolves on a quality level (from first to fifth-Grade) [15–17] and then on a speed level (handwriting speed mainly evolves starting from the fourth grade) [12, 18]. Interestingly, a gender effect has been observed in handwriting acquisition, with girls presenting slightly higher quality and speed scores versus their male peers [17], although no effect of handedness has been reported thus far.

Despite education exposure, 5% to 10% of children never reach a sufficient level of automation in handwriting [17, 19]. These handwriting difficulties, termed dysgraphia, affect legibility and/or speed and can seriously impact both children's behavioural and academical development [6]. As they encounter trouble with automatizing their handwriting, children quickly cannot adequately keep up with the rising cognitive demands of school work, leading to an increase in fatigue and a decrease in cognitive performance. Hence, in children with dysgraphia, the low performance in writing induces negative comparisons with others and self-criticizing. As a consequence, it increases school-performance anxiety and can lead to an increased trait anxiety that can persist until adolescence particularly in boys. This usually generates avoidance of scholarly written tasks, and may eventually result in increased anxiety and low self-esteem, culminating in vicious circles of increasingly fewer writing training opportunities and ultimately school refusal (see Fig 1).

Given the prerequisites of handwriting acquisition, dysgraphia can be related to language problems, motor learning and/or motor execution, visual-motor problems, coordination problems, or cognitive impairments (e.g., attention deficit). In consequence, dysgraphia can be observed in the context of various disorders such as dyslexia, developmental coordination disorders, or attention deficit disorders with or without hyperactivity (ADHD) [20]. Dysgraphia

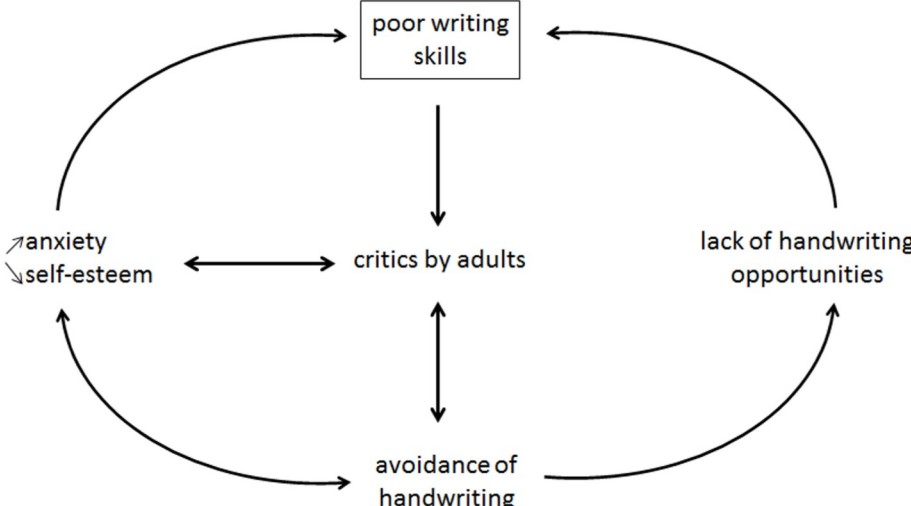

**Fig 1. Psychopathological model of dysgraphia.** Vicious circles can appear due to anxiety and lack of practice that can worsen handwriting. Adapted from [20].

is not recognised by the Diagnostic and Statistical Manual of Mental Disorders, fifth edition (DSM-5) [21] or the International Classification of Diseases 11th edition (ICD-11) [22] as a disorder per se, but can be a specifier of neurodevelopmental disorders. Most classifications of dysgraphia suggest three sub-groups and are usually based on comorbidities. For example, Deuel [23] proposed to differentiate: (1) dyslexic dysgraphia that is often comorbid with attention-deficit or dyslexia; (2) spatial dysgraphia that is the consequence of a defect in the understanding of space; and (3) motor dysgraphia that is often comorbid with a DSM-5 motor acquisition disorder.

The presence of dysgraphia can be assessed via different tests in different alphabets [24]. Concerning the Latin alphabet, we can use the Detailed Assessment of Speed of Handwriting (DASH) [25]; the Ajuriaguerra scale (E scale) [8]; and the Concise Evaluation Scale for Children's Handwriting (BHK) which is the gold-standard test in France for diagnosing dysgraphia. Initially developed in the Netherlands [16], the BHK has since been adapted for use in other languages including French (Charles et al [17]). Importantly, as all of these tests are conducted using a pen/pencil and paper, their scoring is restricted to the analysis of the final, static handwriting product and does not consider or include any information about the movement dynamics.

However, the rising availability of digital tablets in the last few decades has allowed for the analysis of new aspects of handwriting such as the dynamics of handwriting (e.g., velocity, acceleration, etc.) and the pressure of the pen or the pen tilt aspects, for example [24, 26, 27].

Other studies have used digital tablets to better understand handwriting acquisition. Pagliarini et al. [28] employed digital tablets to collect data on handwriting ability before handwriting is performed automatically. The use of quantitative methods enabled them to find patterns indicating potential future writing impairments at a very early age. Separately, Mekyska et al. [29] used a random forest model to classify children with dysgraphia in the Hebrew alphabet. The study included 54 third-grade Israeli children who had to write one Hebrew letter repeatedly. They used static, kinematic, pressure, and tilt aspects of their handwriting as an input to identify poor writing with an excellent accuracy.

In a more recent work, we developed a new test based on the analysis of a large sample (n = 298) of BHKs from typically developing (TD) children and children with dysgraphia [24] written on a digital tablet. This new test allowed for the extraction of 53 digital handwriting features describing several handwriting aspects in the context of four different categories (i.e., statics, kinematics, pressure, and tilt) which were used to train a random forest classifier to be able to diagnose the presence or the absence of dysgraphia with a sensitivity of 96.6% and a specificity of 99.2%.

Despite the inherent progressive learning of writing, so far no study using digital features took into account age and had a developmental approach. We still do not know how the selected features classifying children with dysgraphia evolved in TD children. In addition, we don't know whether their ability to detect children with dysgraphia changed with age.

In the current study, we aimed to extend our work [24] addressing the effect of age, and the heterogeneity of dysgraphia. Our objectives were the following:

1. First, we aimed to present the learning and acquisition of handwriting from a developmental approach (according to child age). We explored TD children in order to better understand typical development (TD dataset only) and children with dysgraphia (D dataset).

2. Second, we aimed to identify the best features, to diagnose children with dysgraphia (according to age) both using the clinical gold standard method as well as relevant digital features [24].

3. Third, we performed unsupervised clustering of children with dysgraphia by applying a K means clustering of discriminative digital features, to assess how many clusters of patients had a similar profile and to identify their main characteristics.

# 1 Materials and methods

## 1.1 Participants

The present study was conducted in accordance with the Declaration of Helsinki and was approved by the Grenoble University Ethics Committee (agreement no. 2016-01-05-79). It was conducted with the understanding and informed written consent of each child's parents and the oral consent of each child. In total, we recruited 280 children. Two hundred thirty-one children were recruited at different schools from Grenoble area. The exclusion criteria were: having a known specific disability or characterized disorder like any neurodevelopmental disorder and being a non-French native. In this study no specific neurological and cognitive assessments were conducted. The absence of disorders was assumed using the teachers' judgments of children's academic achievement. Forty-nine children were recruited on the basis of a clinical diagnosis of dysgraphia from the Reference Center for Language and Learning Disorders at Grenoble University Hospital, a specialized clinic for learning impairments. Since the diagnosis of dysgraphia is not recommended during the 1$^{st}$ grade the children with dysgraphia from this specialized center were excluded. The diagnosis of children from the specialized clinics are reported in Table 1.

**Table 1. Diagnosis of children in the specialized clinic.**

| Isolated disorder: n = 26 (55,3%) | n= |
|---|---|
| DCD | 5 |
| DL | 9 |
| ADHD | 7 |
| dyscalc | 3 |
| dysph | 2 |
| 2 comorbidiess: n = 15 (31,9%) | |
| DCD/DL | 1 |
| DCD/ADHD | 2 |
| DCD/dysph | 2 |
| DL/ADHD | 4 |
| DL/dyscalc | 1 |
| DL/dysph | 2 |
| ADHD/dysph | 2 |
| dyscalc/dysph | 1 |
| 3 comorbidities: n = 5 (10,7%) | |
| DCD/DL/ADHD | 2 |
| DCD/ADHD/dyscalc | 1 |
| DL/ADHD/dysph | 1 |
| DCD/DL/executive disorder | 1 |
| 4 comorbidities: n = 1 (2,1%) | |
| DCD/DL/ADHD/dysph | 1 |

DCD: developmental coordination disorder, DL: dyslexia, ADHD: Attention Deficit Hyperactivity Disorder, dyscalc: dyscalculia, dysph: dysphasia

## 1.2 Procedure

The BHK test consists of copying a text beginning with simple monosyllabic words and evolving towards more complex words for five minutes onto a blank paper. Different features reflecting handwriting quality (e.g., letter form, size, alignment, spacing. . .) are scored to generate a final handwriting quality score. The final handwriting quality score is a degradation score. Higher scores correspond to more errors and a worse quality. A speed score is also provided (i.e., the number of characters written in five minutes) (see BHK scores Table 2).

The 280 children involved in this study performed the BHK test by writing on a sheet of paper affixed to a Wacom graphic tablet (sampling frequency = 200Hz; spatial resolution = 0.25mm). A Wacom Intuos 4 tablet was used for the children recruited in schools, while a

**Table 2. Clinical-Gold Standard (BHK scores) and digital features on handwriting.**

| | | |
|---|---|---|
| **BHK scores** | BHK handwriting quality score based on the sum of 13 quality item scores (raw and normalized with age) | Writing is too large |
| | | Widening of left-hand margin |
| | | Bad letter or word alignment |
| | | Insufficient word spacing |
| | | Chaotic writing |
| | | Absence of joins |
| | | Collision of letters |
| | | Inconsistent letter size (of x-height letters) |
| | | Incorrect relative height of the various kinds of letters |
| | | Letter distortion |
| | | Ambiguous letter forms |
| | | Correction of letter forms |
| | | Unsteady writing trace |
| | BHK speed (raw and normalized with age) | The numbers of characters written in 5 min |
| **Digital features** | Static features | Space between Words |
| | | Standard deviation of handwriting density |
| | | Median of Power Spectral of Tremor Frequencies |
| | Kinematics features | Median of Power Spectral of Speed Frequencies |
| | | Distance to Mean of Speed Frequencies |
| | | In-Air-Time ratio |
| | Pressure features | Mean Pressure |
| | | Mean speed of pressure change |
| | | Standard deviation of speed of pressure change |
| | Tilt features | Distance to Mean of Tilt-x Frequencies |
| | | bandwidth_tiltx |
| | | Median of Power Spectral of Tilt-y Frequencies |

BHK handwriting quality scores have each a score between 0 and 5 according to (1) their age for size of writing and widening of left-hand margin and (2) a score of 0 or 1 for each line of the 1st paragraph for other quality items.

Wacom Intuos 3 tablet was used for the children recruited in the specialized clinic. Pressure data were carefully calibrated between the two tablets. The weights X were carefully chosen (from 0g (pen only) to 400g) in order to explore all range of tablet outputs until saturation. The relation between the weight in input (X) and the value returned by the tablet (Y) could be extracted and was found to be very similar for the two tablets (Spearman correlation > 0.99, $p < 0.001$, mean square error = 0.6). A 4th degree polynomial fit was created to model the function describing the X/Y relation of the first tablet and used on the second to correct the output. After this correction, the spearman correlation was found to be 0.99998 ($p \ll 0.001$) and the mean squared error was 5.1x10-3.

Two junior psychomotor therapists were trained by the same senior psychomotor therapist to score BHK. Then, the 2 juniors therapists annotated independently all BHK both for handwriting quality and speed scores. For the 30 least consistent scores (BHK score >5), the senior therapist scored the BHK.

These professionals were blinded to the demographics and clinical characteristics of the children. Scoring included two dimensions: (1) handwriting velocity assessed through the number of characters written in five minutes and (2) handwriting quality on the five first sentences of the text according to 13 items using a semiquantitative method (BHK handwriting quality scores: Table 2). We calculated the final inter rater-reliability using intra-class correlation, ICC = 0.97 (95% CI: 0.96-0.98). Finally, according to the normal scores by age measured during the previous validation of the scale [17], we computed a qualitative score (quality of the writing) and a quantitative score (speed of the writing).

In a previous work [24], 53 digital handwriting features were defined and used to train a random forest classifier to diagnose dysgraphia. In this work, we only used the features that were found to be the most important in the aforementioned random forest model according to the Gini importance metric [24]. This means that all the features were significantly different between TD and D based on a binary diagnostic classification (BHK threshold). As expected, all digital features were significantly associated with continuous BHK handwriting quality score when the models were applied on TD children and children with dysgraphia. To maintain a good balance and to compare the different groups of features, we selected the three most important features for each of the following four groups that we distinguished: static, pressure, kinematic, and tilt. In the following paragraphs, we briefly provide their respective definitions (Table 2).

**1.2.1 Static features.** They are purely geometric characteristics of the written text. Among static features, we selected: (1) Space Between Words, which refers to the distance between words averaged for the entire text; (2) SD of handwriting density, where a grid with 300-pixel cells covering the entire range of the handwriting trace is created. The number of points recorded by the Wacom tablet in each cell, if present, was stored in an array. The SD of this array is represented by this feature. Also, (3) Median of Power Spectral of Tremor Frequencies was included. Here, the tremors present in the handwriting of children can be calculated for a given packet of points and can thus be described as a series. By doing so, we can apply the usual time series analysis and, in particular, the Fourier transform and take the median of the spectral distribution resulting from it. What we can observe from this is that children having handwriting difficulties show abnormal movements that translate in high frequencies in the Fourier transform, resulting in a shift of the median towards higher frequencies.

**1.2.2 Kinematic features.** They regroup features describing the dynamic of the handwriting process. Among these features, we selected: (1) Median of Power Spectral of Speed Frequencies. We can interpret handwriting as a two-dimensional time series. In the same way as for the Median of Power Spectral of Tremors Frequencies, a Fourier transform can be calculated as well as the median of the spectral distribution resulting from it. We can observe very

fast changes of speed in the handwriting of children with dysgraphia. These abnormal changes of speed are translated in high frequencies in the Fourier transform resulting in a shift of the median towards higher frequencies. (2) Distance to Mean of Speed Frequencies: This feature refers to the distance between the spectral distribution of the writing of the child under investigation and the writing of the typical child of the same age. The higher this distance is, the more eclectic the handwriting of this particular writer is. (3) In-Air-Time ratio: represents the proportion of time spent by the writer without touching the surface of the tablet.

**1.2.3 Pressure features.**   They regroup features using the notion of pressure measured between the pen tip and the tablet surface. Among these features, we selected: (1) Mean Pressure, which is simply the average of all record points of pressure during the test's duration and (2) Mean Speed of Pressure Change, which was extracted by working with averaged buckets of 10 record points of pressure and dividing the time spent by the difference between these two averaged bins of points. This feature is then computed by taking the mean of all measurements. Also, (3) SD of Speed of Pressure Change was selected. This feature is computed in the same way as the feature above: although, instead of applying the mean function, we applied the SD to compute it.

**1.2.4 Tilt features.**   They regroup features using the notion of tilt between the pen and the surface of the tablet. Among these tilt features, we selected: (1) Distance to Mean of Tilt-x Frequencies: This feature refers to the distance between the spectral distribution of the writing of the child under investigation and the one from a typical child of the same age. The higher this distance is, the more eclectic the handwriting of this particular writer. Also, we selected (2) The Bandwidth of Speed of Tilt-x Frequencies: In the same way as described above, the Fourier transform of the two-dimensional time series can be calculated with the tilt-x logs as well as the bandwidth of the spectral distribution resulting from it. For the tilt-x, we can observe that children having handwriting difficulties present spreader tilt-x frequencies and thus a larger bandwidth. Lastly, we included (3) Median of Power Spectral of Tilt-y Frequencies. Here, the Fourier transform of the two-dimensional time series can be calculated with the tilt-y logs as well as the median of the spectral distribution resulting from it. For the tilt-y, we can observe that children having handwriting difficulties present lower tilt-y frequencies and thus a lower median.

**1.2.5 Statistical models.**   Since we selected the 12 digital features through machine learning classifying BHK scores as threshold (binary classification), we considered inappropriate to use direct group comparisons (TD vs. D). To take into account the effect of age, and possible interactions between a given digital feature and age, we applied linear regressions considering each feature as a continuous variable to explain the BHK as a continuous variable without consideration of the diagnosis threshold. To do so, each of the 12 digital features was normalized in order to assess their effect in linear regression models.

To understand how a given digital feature is explaining or not BHK handwriting quality taking into account grade and gender, a linear regression model per feature was created to predict the continuous BHK handwriting quality score. This model was adjusted for grade and gender. In the same way, a model was created to predict the BHK speed score. The formulas can be described as follows:

$$BHK\ Score \sim Normalized(feature) + grade + gender + \varepsilon$$

To understand how a given digital feature explaining continuous BHK changes according to a child's grade, a similar model with interaction [grade*Normalized(feature)] was also created. In other words, the model can show the relative importance of a given digital feature to diagnose dysgraphia according to age. As recommended in the BHK manual, we selected the

grade rather than the age to assess the effect on education, since the writing process is learned at school and not spontaneously. The formulas can be described as follows:

$$BHK\ Score \sim Normalized(feature) + grade + gender+$$
$$+grade * Normalized(feature) + \varepsilon \tag{1}$$

Since the distribution of the residuals was not normally distributed, a bootstrap analysis (with 10,000 replications) was performed to assess the 95% confidence intervals (95%CI) and p values. These were respectively obtained by BCA (bias-corrected and accelerated) bootstrap and percentile bootstrap with the R boot package. As said previously, we performed these analyses on the TD dataset only to explore how digital features predict handwriting (BHK) quality and speed in TD children, then on the TD + D dataset to explore how digital features predict handwriting (BHK) quality and speed in a mixed population that resembles a more realistic situation in the context of school detection of D children.

**1.2.6 Clustering.** Finally, we tested the theoretical classification of Deuel [23] by a K-means clustering of our digital features, to assess how many clusters of patients had a similar profile and to identify their main characteristics. We used the elbow method to explore the best numbers of clusters.

## 2 Results

### 2.1 Participant's demographics

Our first aim was to better characterize the children recruited from schools and to assess whether or not few had dysgraphia. After clinical assessment of BHK tests of the 280 children from our dataset, we confirmed dysgraphia in all children recruited in the special clinics and detected 13 (5.63%) children with dysgraphia among those recruited from regular schools. Speed dysgraphia (slow writing) was observed in 12 children, with all of them showing also qualitative dysgraphia (poor handwriting quality, legibility). Thus, we defined the diagnostic category of dysgraphia based on the BHK handwriting quality score only. Therefore, after re-annotation of all BHKs, our dataset was composed of 218 children in the control group, without dysgraphia called the TD group, and 62 children in the experimental group, with dysgraphia called the D group. (see Fig 2).

Table 3 summarizes the main characteristics of the two groups. The TD and D children had similar ages of around nine years on average despite a tendency of older age in the group with dysgraphia. Most children were right-handed. There was a gender bias (girls were underrepresented in the D group).

### 2.2 Handwriting acquisition

Our second aim was to identify the best features to diagnose children with dysgraphia both using BHK scores (the clinical gold standard method) and relevant digital features, and to explore how the relevant digital features had statistical interaction with age.

**2.2.1 Handwriting explained from the BHK features.** Fig 3 summarizes the handwriting quality and speed BHK scores for both the TD and D datasets. As expected, we could see an improvement of the handwriting quality (decrease of the BHK quality score) together with an increase of the writing speed with the age of children. By definition, children with dysgraphia had a lower quality versus TD children. Normalized score allow comparisons between grades. The cut-off for a diagnostic of qualitative and quantitative dysgraphia is -2.

**2.2.2 Handwriting explained from the digital features.** 12 digital features expressing handwriting on different aspects (static, kinematics, pressure, and tilt) were selected from the

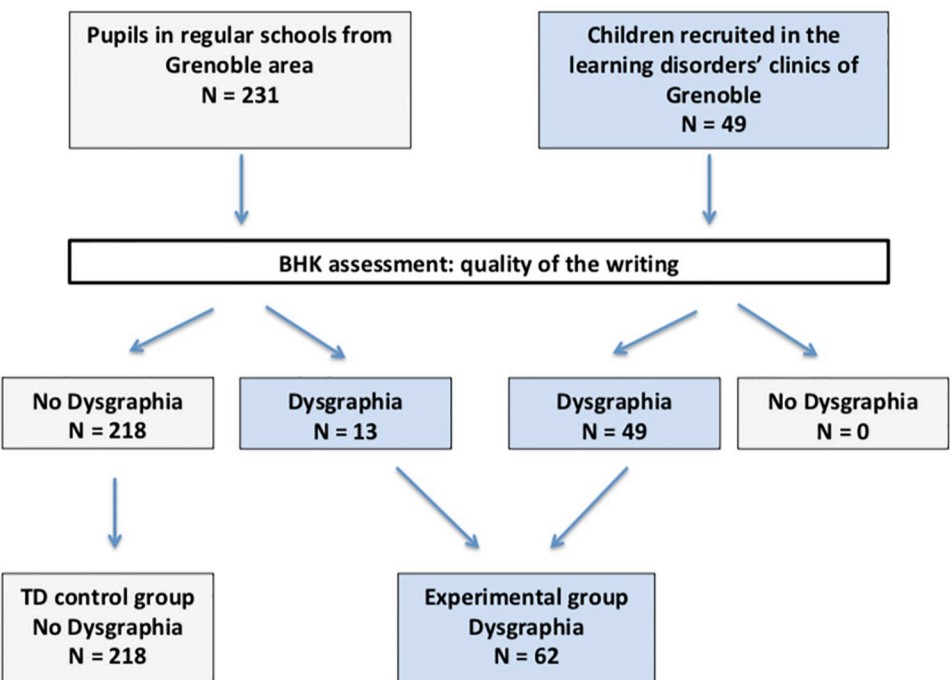

**Fig 2. Annotation of the database with the BHK test defining children with dysgraphia (writing quality too bad, BHK handwriting quality score too high) and children without dysgraphia.**

**Table 3. Descriptive statistics of the participants (TD and D).**

| | TD group (No dysgraphia) (n = 218) | D group (Dysgraphia) (n = 62) | p-value |
|---|---|---|---|
| **Age**: mean (SD) | 8.7 (1.53) | 9.13 (1.2) | 0.056 |
| **Males/Females** | 108/110 | 44/18 | **0.003** |
| **Right-handed / Left-handed** | 190/28 | 57/5 | 0.30 |
| BHK quality score: mean (SD) | 14.41 (5.16) | 27.09 (6.83) | **0.001** |
| Grade 1: mean (SD) | n = 48; 20.07 (5.39) | n = 1; 35.5 | 0.10 |
| Grade 2: mean (SD) | n = 42; 13.17 (4.56) | n = 16;33.35 (6.22) | **<0.001** |
| Grade 3: mean (SD) | n = 36; 13.6 (3.66) | n = 15; 26.99 (5.76) | **<0.001** |
| Grade 4: mean (SD) | n = 44; 12.68 (3.4) | n = 20; 24.1 (4.75) | **<0.001** |
| Grade 5: mean (SD) | n = 48; 12.02 (3.45) | n = 10; 22.37 (5.51) | **<0.001** |
| BHK speed scores: mean (SD) | 195.9 (94.6) | 139.5 (86.6) | **<0.001** |
| Grade 1: mean (SD) | n = 48; 74.83 (18.32) | n = 1; 61,5 | 0.52 |
| Grade 2: mean (SD) | n = 42; 152.29 (43.27) | n = 16; 70.4 (23.53) | **<0.001** |
| Grade 3: mean (SD) | n = 36; 184.97 (42.43) | n = 15; 123.67 (47.36) | **<0.001** |
| Grade 4: mean (SD) | n = 44; 262.27 (56.99) | n = 20; 160.78 (81.62) | **<0.001** |
| Grade 5: mean (SD) | n = 48; 302.54 (50.42) | n = 10; 239.28 (103.53) | **0.03** |

Non parametric (Wicoxon rank sum) tests were performed to compare BHK quality score for the 2 whole datasets and for each grade, (SD for standard deviation, TD for Typically Developing, D for dysgraphia)

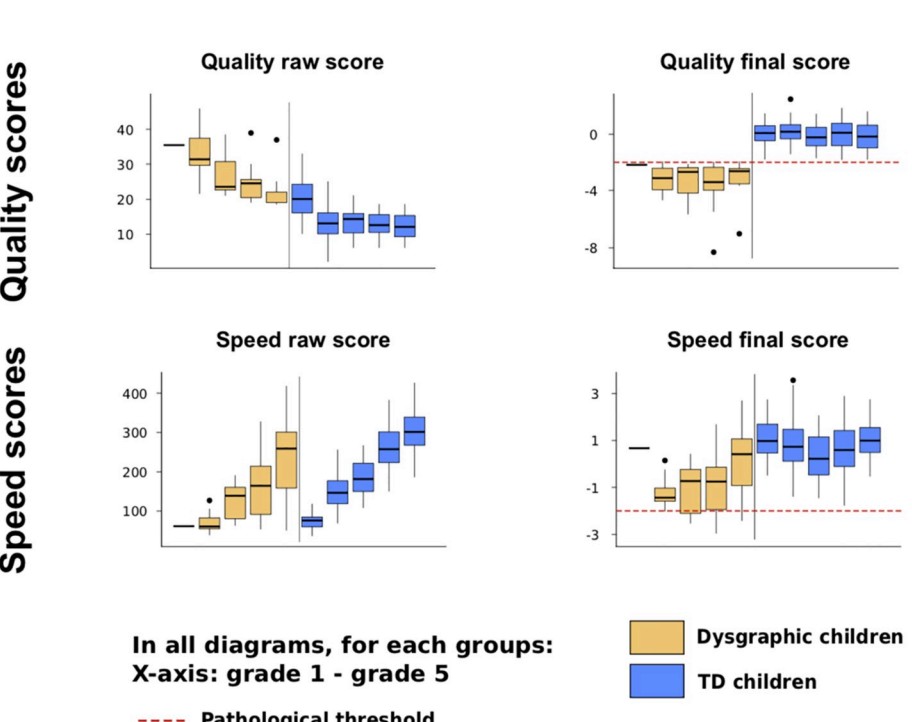

**Fig 3. BHK handwriting quality and speed scores according to grade in children with typical development and in children with dysgraphia.** Raw score (left) and normalized score (right). Notice that the BHK handwriting quality score is a degradation score. The higher the score is, the more the handwriting is impaired. The speed raw score is the number of letters written during 5 minutes.

work of Asselborn et al. [24]. In Fig 4, the link between the digital features and the BHK raw handwriting quality score in terms of function of the grade is presented (children without dysgraphia, TD dataset).

Since we selected these features on the basis of their importance on the simple binary classification between children with or without dysgraphia, we wanted to assess whether they were also able to explain the continuous BHK handwriting quality score (inverse of the handwriting writing quality) of both the D dataset and TD datasets. For this purpose, multivariate linear regression models were used to compute the correlation between the 12 digital features and the BHK scores (handwriting quality or speed).

**Handwriting quality score association.** Table 4 shows the association between the digital features and the raw BHK handwriting quality score (opposite of handwriting quality) for the children without dysgraphia only (TD dataset) as well as for all children taken altogether (D and TD datasets). As expected, since all these features already classified in a proper manner dysgraphia on a simple binary classification in Asselborn et al. [24], all digital features were also significantly associated with continuous BHK handwriting quality score when the models were applied on TD children and children with dysgraphia. However, only three digital features (out of 12) were significantly associated with the BHK handwriting quality score of TD children.

This finding is interesting, as it shows that the digital features seem to belong to two groups: all of them are useful to explain the difference of handwriting quality between TD children

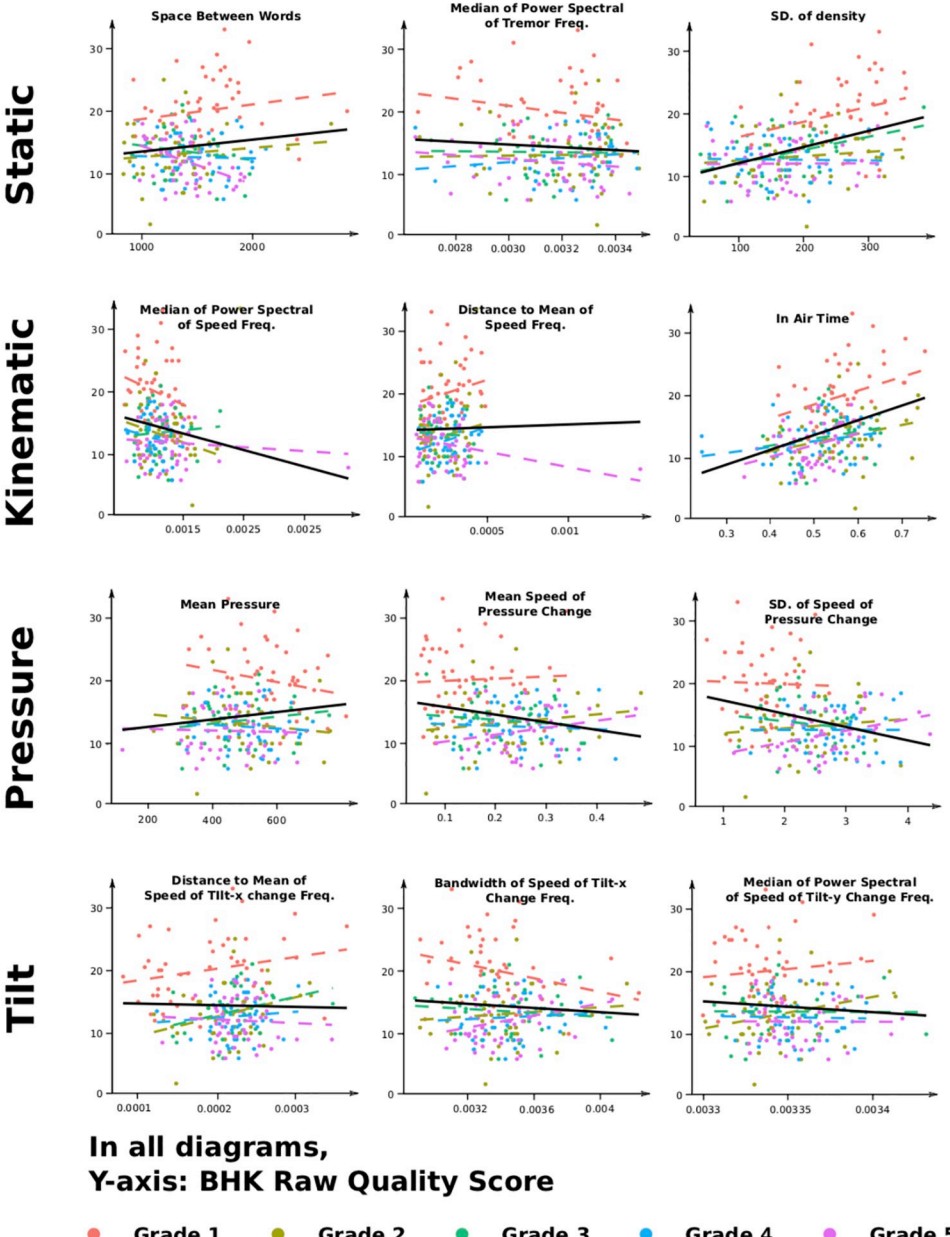

**Fig 4. The impact of the 12 digital features on the BHK raw handwriting quality score (opposite of handwriting quality) for children without dysgraphia (TD dataset) and for all grades.**

and children with dysgraphia, but only a few features are useful for predicting the handwriting quality of TD children without dysgraphia. In other words, it means that some features become interesting to predict the handwriting quality score only when the score is above a certain threshold (i.e., the score only reached by children with dysgraphia as can be seen in Fig 3). These features are the ones able to explain the handwriting quality difference between children with dysgraphia and TD children.

As there are several potential causes of dysgraphia with different severities, the variability of handwriting is larger in the D dataset versus the TD dataset. In view of the limited size of our

**Table 4. Multivariate models to predict the BHK handwriting quality raw score.**

| Category | Feature | Dataset | Feature estimate | Cl95% low | Cl95% up | p value |
|---|---|---|---|---|---|---|
| **Static** | Space Between Words | TD | 0.45 | -0.11 | 1.05 | 0.116 |
| | | TD + D | -1.16 | -2.15 | 0.008 | 0.033 |
| | SD of Handwriting Density | TD | 0.98 | 0.34 | 1.67 | 0.004 |
| | | TD + D | 2.09 | 1.03 | 3.04 | <0.001 |
| | Median of Power Spectral of Tremor Freq. | TD | -0.25 | -0.8 | 0.31 | 0.374 |
| | | TD + D | 1.39 | 0.68 | 2.19 | 0.001 |
| **Kinematic** | Median of Power Spectral of Speed Freq. | TD | -0.48 | -1.13 | -0.03 | 0.038 |
| | | TD + D | 3.65 | 2.68 | 4.49 | <0.001 |
| | Distance to mean of Speed Freq. | TD | 0.17 | -0.29 | 0.9 | 0.479 |
| | | TD + D | 2.42 | 1.43 | 3.45 | <0.001 |
| | In Air Time Ratio | TD | 0.85 | 0.23 | 1.52 | 0.006 |
| | | TD + D | 2.44 | 1.36 | 3.41 | <0.001 |
| **Pressure** | Mean Pressure | TD | -0.03 | -0.65 | 0.57 | 0.935 |
| | | TD + D | -1.25 | -2.07 | -0.3 | 0.008 |
| | Mean Speed of Pressure Change | TD | 0.09 | -0.58 | 0.84 | 0.793 |
| | | TD + D | -2.56 | -3.51 | -1.61 | <0.001 |
| | SD of speed of Pressure Change | TD | -0.12 | -0.9 | 0.61 | 0.769 |
| | | TD + D | -2.32 | -3.3 | -1.4 | <0.001 |
| **Tilt** | Distance to Mean of Tilt-x Freq. | TD | 0.42 | -0.21 | 1.03 | 0.178 |
| | | TD + D | 3.35 | 2.54 | 4.1 | <0.001 |
| | Bandwidth of Power Spectral of Tilt-x Freq. | TD | -0.22 | -0.78 | 0.31 | 0.435 |
| | | TD + D | 2.2 | 1.17 | 3.06 | <0.001 |
| | Median of Power Spectral of Tilt-y Freq. | TD | 0.16 | -0.45 | 0.76 | 0.597 |
| | | TD + D | -2.46 | -3.27 | -1.65 | <0.001 |

The higher the score, the lower the handwriting quality, for typically developing children only (TD dataset), and all children together (TD + D dataset).

D dataset, the interpretation of the digital feature acquisition of the children with dysgraphia may be difficult. For this reason, we decided to focus our feature interpretation on the children without dysgraphia.

Among the three features able to explain the BHK handwriting quality score of children without dysgraphia only, we found two kinematic features (Median Power Spectral Speed Frequencies and In Air Time, respectively) and one static feature (Standard Deviation (SD) of Handwriting Density). As can be seen in Table 4 and in Fig 4, the In Air Time Ratio as well as the SD of Handwriting Density are positively correlated with the BHK handwriting quality score. This means that a reduction of the In Air Time Ratio or the SD of Handwriting density is with an increase in the quality of handwriting (i.e., the lower the score is, the better the quality is). Separately, the Median Power Spectral of Speed Frequency was negatively correlated with the BHK handwriting quality score-in other words, the higher the value of this feature is, the better the handwriting quality is.

**Handwriting speed association.** Concerning the BHK speed score, the same three features were significantly correlated with the BHK speed score of TD children.

A fourth one namely, the SD of Speed of Pressure Change, was also found to be significantly associated (see Table 5). The Median of Power Spectral of Speed Frequencies was found to be negatively correlated with the BHK speed score, meaning that, the lower this feature is (indicating a less brutal change in handwriting speed), the higher the speed of handwriting will be.

**Table 5. Multivariate models to predict the BHK speed raw score for typically developing children (TD dataset), and all children together (TD + D dataset).**

| Category | Feature | Dataset | Feature estimate | Cl95% low | Cl95% up | p value |
|---|---|---|---|---|---|---|
| **Static** | Space Between Words | TD | 5.55 | 0.01 | 11.46 | 0.057 |
| | | TD + D | 16.6 | 5.64 | 25.1 | **<0.001** |
| | SD of Handwriting Density | TD | -16.89 | -23.5 | -9.53 | **<0.001** |
| | | TD + D | -21.18 | -29.19 | -12.56 | **<0.001** |
| | Median of Power Spectral of Tremor Freq. | TD | 0.62 | -4.76 | 6.05 | 0.831 |
| | | TD + D | -12.14 | -18.47 | -6.28 | **<0.001** |
| **Kinematic** | Median of Power Spectral of Speed Freq. | TD | -6.24 | -12.94 | -1.32 | 0.013 |
| | | TD + D | -31.37 | -38.65 | -24.26 | **<0.001** |
| | Distance to mean of Speed Freq. | TD | 4.15 | -1.39 | 12.1 | 0.09 |
| | | TD + D | -19.82 | -27.34 | -13.34 | **<0.001** |
| | In Air Time Ratio | TD | -21.37 | -27.6 | -14.85 | **<0.001** |
| | | TD + D | -27.71 | -35.66 | -19.62 | **<0.001** |
| **Pressure** | Mean Pressure | TD | 4.91 | -0.66 | 10.54 | 0.093 |
| | | TD + D | 9.13 | 1.67 | 16.33 | **0.012** |
| | Mean Speed of Pressure Change | TD | 5.43 | -1.03 | 11.92 | 0.09 |
| | | TD + D | 23.45 | 15.66 | 31.55 | **<0.001** |
| | SD of speed of Pressure Change | TD | 17.81 | 11.35 | 24.28 | **<0.001** |
| | | TD + D | 29.96 | 22.09 | 37.89 | **<0.001** |
| **Tilt** | Distance to Mean of Tilt-x Freq. | TD | -1.84 | -6.21 | 2.87 | 0.424 |
| | | TD + D | -18.76 | -24.55 | -13.49 | **<0.001** |
| | Bandwidth of Power Spectral of Tilt-x Freq. | TD | 0.31 | -5.89 | 6.87 | 0.929 |
| | | TD + D | -12.96 | -19.65 | -6.26 | **<0.001** |
| | Median of Power Spectral of Tilt-y Freq. | TD | 4.41 | -1.23 | 10.52 | 0.118 |
| | | TD + D | 19.59 | 13.96 | 25.94 | **<0.001** |

In the same way, the In Air Time Ratio was found to be negatively correlated with the handwriting speed (a reduction in the time the child has the pen not touching the paper will result in more time spent writing and thus a higher handwriting speed) as well as the SD of Handwriting density (which can be interpreted as the fluctuation in the handwriting size). Finally, the SD of Speed of Pressure Change was found to be positively correlated with the handwriting speed. Clinically, this feature can be linked with the automation of the pen movement (i.e., the child has a better control of the pen if he/she is able to change the pressure of the pen in different ways).

**Interaction of the digital features with grade.** Considering a developmental analysis of handwriting, we used multivariate models to predict children's BHK scores (quality and speed), taking into account the digital feature, the grade, the gender, but also the interaction between the feature and the grade. Results of the interaction are reported in Table 6 for the BHK handwriting quality score and in Table 7 for the BHK speed score, respectively.

Quality interaction: Once again, we can see that the interaction between handwriting digital features and grade seems to be different between children with dysgraphia and TD children. If we consider the features of the TD children alone, eight of the 12 features present a statistically significant interaction with grade to predict BHK handwriting quality, while only one (Space Between Words) is significant if we add the children with dysgraphia to the dataset on which the model is applied (Table 6). We believe that this can be explained by the fact that children with dysgraphia show a very different handwriting manner (and thus very different digital features) and present a significantly higher heterogeneity in their writing (and thus more spread-

**Table 6. Multivariate models with interaction to predict the BHK quality raw score.**

| Category | Feature | Dataset | Feature * grade estimate | Cl95% low | Cl95% up | p value |
|----------|---------|---------|--------------------------|-----------|----------|---------|
| **Static** | Space Between Words | TD | -0.72 | -1.18 | -0.31 | **0.001** |
| | | TD + D | -1 | -1.62 | -0.38 | **0.001** |
| | SD of Handwriting Density | TD | -0.77 | -1.2 | -0.35 | **0.001** |
| | | TD + D | -0.44 | -1.06 | 0.26 | 0.197 |
| | Median of Power Spectral of Tremor Freq. | TD | 0.22 | -0.17 | 0.6 | 0.237 |
| | | TD + D | 0.16 | -0.35 | 0.63 | 0.527 |
| **Kinematic** | Median of Power Spectral of Speed Freq. | TD | 0.52 | 0.08 | 0.92 | **0.003** |
| | | TD + D | -0.42 | -1.27 | 0.34 | 0.357 |
| | Distance to mean of Speed Freq. | TD | -0.5 | -0.89 | -0.07 | **0.04** |
| | | TD + D | -0.8 | -1.48 | 0.09 | 0.073 |
| | In Air Time Ratio | TD | -0.48 | -0.92 | -0.03 | **0.04** |
| | | TD + D | -0.15 | -0.78 | 0.53 | 0.637 |
| **Pressure** | Mean Pressure | TD | -0.18 | -0.57 | 0.2 | 0.333 |
| | | TD + D | 0.26 | -0.23 | 0.75 | 0.313 |
| | Mean Speed of Pressure Change | TD | 0.54 | 0.04 | 0.94 | **0.023** |
| | | TD + D | 0.45 | -0.29 | 1.14 | 0.216 |
| | SD of speed of Pressure Change | TD | 0.82 | 0.38 | 1.24 | **0.001** |
| | | TD + D | 0.46 | -0.15 | 1.05 | 0.12 |
| **Tilt** | Distance to Mean of Tilt-x Freq. | TD | 0.02 | -0.36 | 0.42 | 0.891 |
| | | TD + D | -0.52 | -1.2 | 0.3 | 0.218 |
| | Bandwidth of Power Spectral of Tilt-x Freq. | TD | 0.43 | 0.03 | 0.84 | **0.028** |
| | | TD + D | -0.11 | -0.81 | 0.59 | 0.774 |
| | Median of Power Spectral of Tilt-y Freq. | TD | -0.05 | -0.53 | 0.44 | 0.852 |
| | | TD + D | -0.09 | -0.76 | 0.58 | 0.755 |

The higher the score, the lower the handwriting quality) for typically developing children (TD dataset) and all children together (TD + D dataset).

out) as compared with TD children, which brings additional noise to the model and avoids us from finding significant interactions.

For the same reasons as before, we decided to focus our interpretation on TD children only. As can be seen in Fig 4 and Table 6 an interaction between most of the digital features with the grade was found to predict the BHK raw handwriting quality score. Only two tilt features (Distance to Mean of Tilt-x Frequencies, Median of Power Spectral of Tilt-y Frequencies), one static feature (Median of Power Spectral of Tremor Frequencies) as well as one pressure feature (Mean Pressure) did not present a statistically significant interaction with the grade. This result shows that the predictive value of most features changes with the grade. In other words, most of them are either useful for detecting the quality of the writing for the younger or for the older children. A typical example of this can be found with the Space Between Words: this feature is positively associated with BHK score in first grade and becomes negatively associated by fifth grade (see Fig 4). In other words, as the child is progressing in his/her school curriculum, the Space Between Words feature becomes more and more so a negative predictor of the BHK quality score (and thus a positive predictor of handwriting quality).

Handwriting speed interaction: Models with interaction between features and grade to predict BHK speed are presented in Table 7. In the TD dataset, two kinematic features (Median of Power Spectral of Speed Frequencies and Distance to Mean of Speed Frequencies) and one pressure feature (mean pressure) showed a significant interaction. When models were applied in TD+D dataset, two kinematic features (Distance to Mean of Speed Frequencies and In Air

**Table 7. Multivariate models with interaction to predict the BHK speed raw score for typically developing children (TD dataset) and all children together (TD + D dataset).**

| Category | Feature | Dataset | Feature * grade estimate | Cl95% low | Cl95% up | p value |
|---|---|---|---|---|---|---|
| **Static** | Space Between Words | TD | 1.24 | -2.81 | 5.33 | 0.52 |
| | | TD + D | 8.64 | 3.25 | 13.96 | **0.003** |
| | SD of Handwriting Density | TD | -0.1 | -4.37 | 4.69 | 0.948 |
| | | TD + D | -3.14 | -8.97 | 3.58 | 0.318 |
| | Median of Power Spectral of Tremor Freq. | TD | 1.22 | -2.21 | 4.87 | 0.514 |
| | | TD + D | -2.47 | -6.66 | 1.65 | 0.203 |
| **Kinematic** | Median of Power Spectral of Speed Freq. | TD | 3.27 | -0.02 | 7.39 | **0.039** |
| | | TD + D | -4.4 | -10.83 | 3.09 | 0.223 |
| | Distance to mean of Speed Freq. | TD | -4.32 | -7.62 | -0.5 | **0.038** |
| | | TD + D | -7.85 | -15.72 | -0.53 | **0.005** |
| | In Air Time Ratio | TD | -2.29 | -6.46 | 2.12 | 0.279 |
| | | TD + D | -9.29 | -14.64 | -3.69 | **0.001** |
| **Pressure** | Mean Pressure | TD | 3.89 | 0.55 | 7.61 | **0.028** |
| | | TD + D | 2 | -2.19 | 6.26 | 0.337 |
| | Mean Speed of Pressure Change | TD | -2.11 | -6.14 | 1.96 | 0.306 |
| | | TD + D | 2.84 | -2.28 | 8.47 | 0.299 |
| | SD of speed of Pressure Change | TD | 1.56 | -2.03 | 5.32 | 0.391 |
| | | TD + D | 5.7 | 1.28 | 10.28 | **0.009** |
| **Tilt** | Distance to Mean of Tilt-x Freq. | TD | -2.52 | -6.27 | 1.83 | 0.206 |
| | | TD + D | -3.64 | -10.46 | 1.68 | 0.209 |
| | Bandwidth of Power Spectral of Tilt-x Freq. | TD | -1.93 | -6.54 | 2.78 | 0.407 |
| | | TD + D | -0.64 | -6.59 | 4.93 | 0.832 |
| | Median of Power Spectral of Tilt-y Freq. | TD | -1.94 | -6.09 | 2.19 | 0.346 |
| | | TD + D | 4.43 | -1.15 | 9.8 | 0.111 |

Ratio), one static feature (Space Between Words) and one Pressure feature (SD of Speed of Pressure Change) showed a significant interaction with grade to predict speed.

Interestingly, Space Between Words is the only feature that interacts with grade to predict both BHK quality and speed in the TD+D dataset. It becomes increasingly positively correlated with the BHK speed score with the increase in the child's grade, which means that, at the beginning of the school curriculum, the space that children put between words is not a significant predictor of handwriting speed, but becomes a stronger one as the child continues in his/her school education.

## 2.3 A new clustering of dysgraphia

Our third aim was to explore whether we could define a new classification of dysgraphia based on our automated features. Hence, most of the classifications of dysgraphia are based on comorbidities (e.g., dyslexia, attention problems, motor acquisition problems). A more precise description of dysgraphia should take into account the peculiar objective characteristics of handwriting. To do this, we used a K-means clustering algorithm with our 12 digital features as input. Using the elbow method to explore the best number of clusters, we found that three clusters was an optimal number according to the majority rule (see Fig 5).

Regarding the final model, the Hopkins statistic was 0.35 and the clusters' stability were satisfactory (cluster 1: 0.87, cluster 2: 0.89 and cluster 3: 0.84). As can be seen in Table 8, individuals from each cluster had the following attributes:

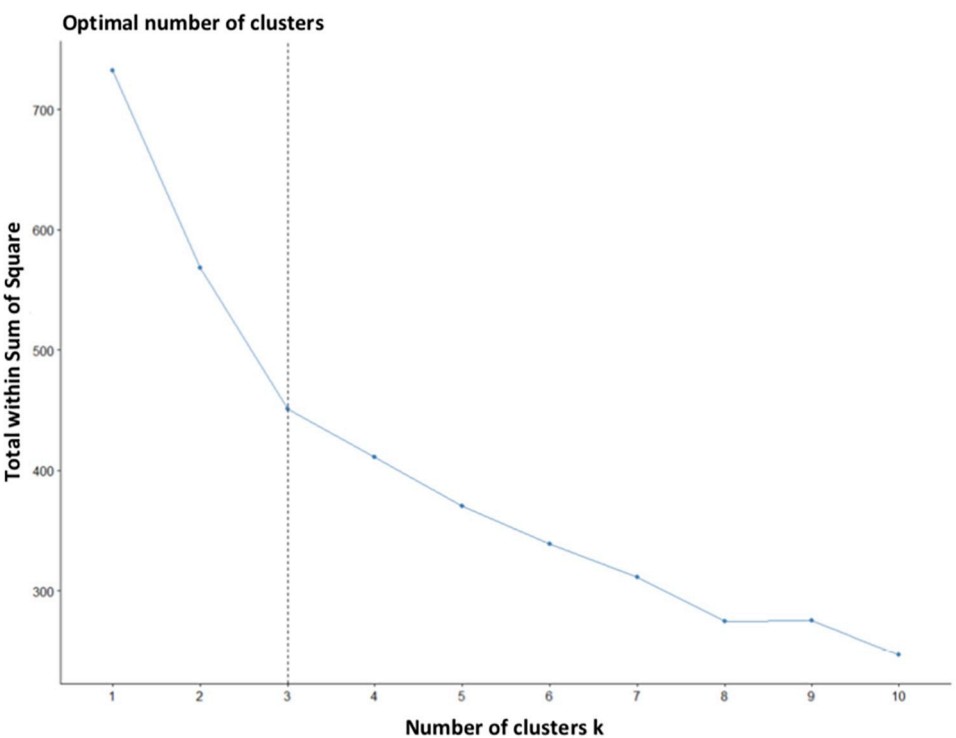

**Fig 5. Elbow method to characterize the optimal number of clusters.**

**Table 8. Mean digital features value of each features according to their clustering.** Demographic characteristics and BHK scores of children with dysgraphia corresponding to the 3 clusters.

| Category | Feature | Cluster 1 (n = 13) | Cluster 2 (n = 25) | Cluster 3 (n = 24) | p-value |
|---|---|---|---|---|---|
| Statics | Space Between Words | $1.3 \cdot 10^3 (3 \cdot 10^2)$ | $7.2 \cdot 10^2 (2.9 \cdot 10^2)$ | $1.4 \cdot 10^3 (6 \cdot 10^2)$ | **<0.001** |
| | Median of Power Spectral of Tremor Freq. | $3.2 \cdot 10^{-3} (1.5 \cdot 10^{-4})$ | $3.3 \cdot 10^{-3} (9.7 \cdot 10^{-5})$ | $3.3 \cdot 10^{-3} (1.2 \cdot 10^{-4})$ | 0.055 |
| | SD of Handwriting Density | $1.7 \cdot 10^2 (73)$ | $2.3 \cdot 10^2 (1.3 \cdot 10^2)$ | $2.5 \cdot 10^2 (1.2 \cdot 10^2)$ | 0.161 |
| Kinematics | Median of Power Spectral of Speed Freq. | $1.4 \cdot 10^{-3} (1.6 \cdot 10^{-4})$ | $2.2 \cdot 10^{-3} (2.8 \cdot 10^{-4})$ | $1.9 \cdot 10^{-3} (2.8 \cdot 10^{-4})$ | **<0.001** |
| | Distance to mean of Speed Freq | $2.1 \cdot 10^{-4} (9.6 \cdot 10^{-5})$ | $7.7 \cdot 10^{-4} (3.5 \cdot 10^{-4})$ | $4.1 \cdot 10^{-4} (2.7 \cdot 10^{-4})$ | **<0.001** |
| | In Air Time Ratio | $5.3 \cdot 10^{-1} (7.8 \cdot 10^{-2})$ | $5.9 \cdot 10^{-1} (1.1 \cdot 10^{-1})$ | $6 \cdot 10^{-1} (1.4 \cdot 10^{-1})$ | 0.104 |
| Pressure | Mean Pressure | $4.6 \cdot 10^2 (1.4 \cdot 10^2)$ | $3.2 \cdot 10^2 (1.1 \cdot 10^2)$ | $4.6 \cdot 10^2 (1.4 \cdot 10^2)$ | **0.001** |
| | Mean Speed of Pressure Change | $3.1 \cdot 10^{-1} (1.3 \cdot 10^{-1})$ | $3.8 \cdot 10^{-2} (3.5 \cdot 10^{-2})$ | $6.5 \cdot 10^{-2} (5.6 \cdot 10^{-2})$ | **<0.001** |
| | SD of speed of Pressure Change | $28 (7.5 \cdot 10^{-1})$ | $13 (4 \cdot 10^{-1})$ | $20 (5.1 \cdot 10^{-1})$ | **<0.001** |
| Tilt | Median of Power Spectral of Tilt-y Freq | $3.3 \cdot 10^{-3} (1.5 \cdot 10^{-5})$ | $3.3 \cdot 10^{-3} (3.1 \cdot 10^{-5})$ | $3.3 \cdot 10^{-3} (5.5 \cdot 10^{-5})$ | **0.001** |
| | Distance to Mean of Tilt-x Freq | $2.3 \cdot 10^{-4} (3.3 \cdot 10^{-5})$ | $2.8 \cdot 10^{-4} (9.6 \cdot 10^{-5})$ | $6.3 \cdot 10^{-4} (1.9 \cdot 10^{-4})$ | **<0.001** |
| | Bandwidth_tilt x Freq. | $3.4 \cdot 10^{-3} (2.8 \cdot 10^{-4})$ | $3.5 \cdot 10^{-3} (3.8 \cdot 10^{-4})$ | $4.4 \cdot 10^{-3} (2.9 \cdot 10^{-4})$ | **<0.001** |
| Clinical description | Grade | 4.08 (0.86) | 3.32 (1.18) | 3.00 (0.93) | **0.016** |
| | Gender (F/M) | 8/5 | 4/21 | 6/18 | **0.016** |
| | Laterality (L/R) | 0/13 | 1/24 | 4/20 | 0.212 |
| | Recruited in Specialized Clinic | 0 | 25 | 24 | **<0.001** |
| | BHK handwriting quality Score | -2.73 (0.65) | -3.49 (1.25) | -3.44 (1.41) | 0.109 |
| | BHK handwriting speed Score | 0.46 (1.16) | -1.24 (1.00) | -1.10 (0.91) | **<0.001** |

The clustering was done on the digital features only

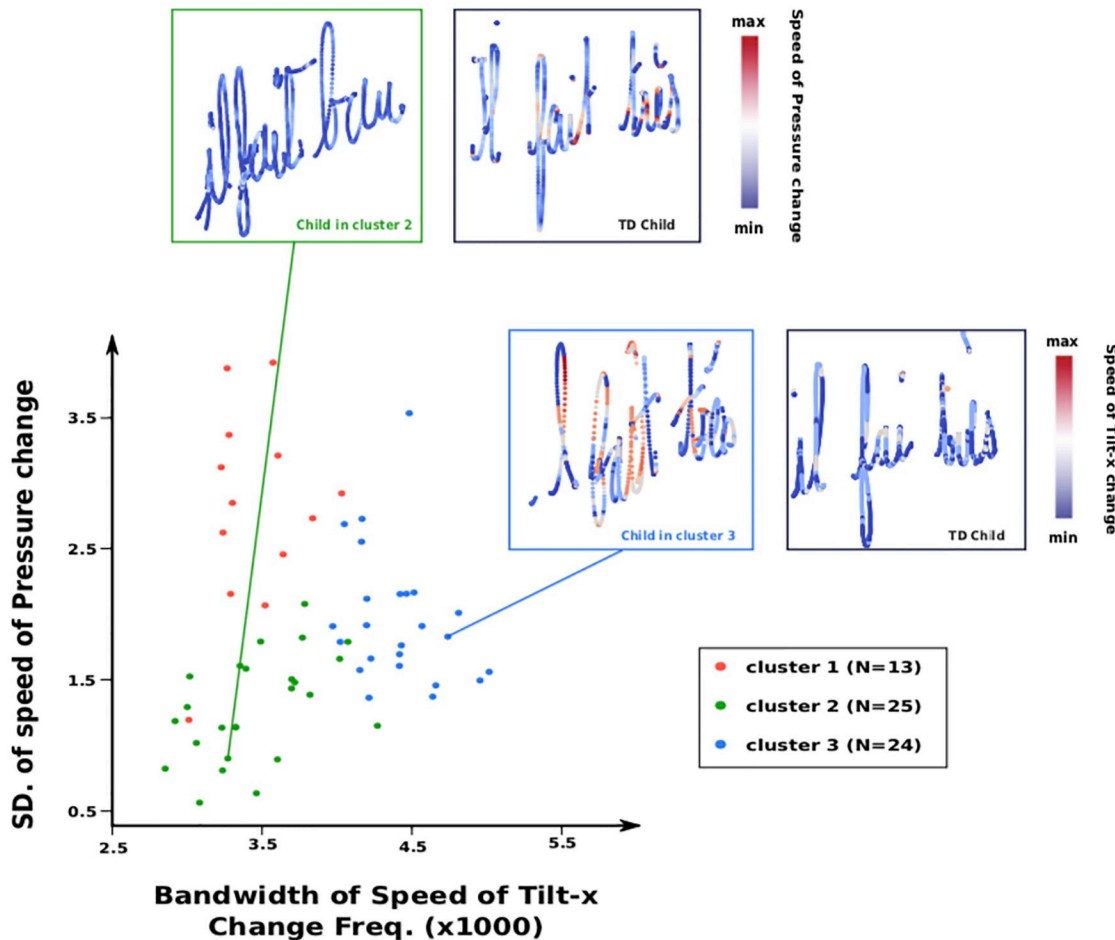

**Fig 6. Comparisons of the SD of speed of pressure change and Bandwidth of Speed of Tilt-x Change Frequencies for the children without dysgraphia from the 3 different clusters.** Examples of writing from a child with the most severe difficulties from cluster 2 and cluster 3 are shown.

Children from cluster 1 (n = 13) presented the less severe type of dysgraphia, with girls and older children being over-represented. Their BHK speed score was significantly higher. Also, they tended to have better BHK handwriting quality normalized score (meaning better handwriting quality) (Table 8). Concerning the digital features, it was interesting to see a significantly lower value (as compared with the other two clusters) of the Median of Power Spectral of Speed Frequencies, meaning that the variation of the handwriting speed is slower for these children (i.e., the transition between low and high speeds is smoother). In addition, the SD of Speed of Pressure Change was found to be significantly higher, also suggesting a better automation of the pen movement. An example of this feature can be found in Fig 6, where we can see that the Speed of pressure Change of a child with dysgraphia (from cluster 2) stays relatively constant as opposite to that in the handwriting example of the TD child.

Children from clusters 2 and 3 were all recruited from a specialized clinic, meaning that they presented more severe cases of dysgraphia. As can be seen in Table 8, a statistically significant majority of these children were boys, which appears to be in line with the findings of previous studies [15, 17]. Concerning the digital features, children from cluster 2 (n = 25) were characterized as having a lower Space Between Words versus the two other clusters and

particular abnormalities in the speed frequencies as well as the pressure features (see Table 8). The Distance to Mean of Speed Frequencies was found to be statistically higher here than in the two other clusters, meaning that these children were the most eclectic, regarding the way their handwriting speed was changing. The accelerations (and decelerations) of the handwriting speed were more abrupt, suggesting a lack of automation in the control of the speed (i.e., more jerk recorded). The other features for which children from cluster 2 were noted to be particularly different were the pressure features. As can be seen in the example of Fig 6, children from cluster 2 were using a much smaller gaps of pressure while writing (i.e., the pressure stays relatively more constant during the handwriting) in comparison with TD children.

Children from cluster 3 (n = 24) were characterized by abnormalities in terms of tilt features. As can be seen in Fig 6, children from this cluster had troubles with smoothly changing the inclination of their pen (i.e., the transition between low and high speeds of tilt change was found to be less constant versus in TD children).

## 3 Discussion

The clinical annotation of our dataset found 13 (5.63%) children with dysgraphia in schools. This appears consistent with the 5%-10% prevalence rates reported in the literature as well in the most recent French study [17]. Among these 13 cases of dysgraphia, one child was currently in the first grade. This aligns with the fact that children who are referred to specialized clinics for dysgraphia are usually older. This could be explained by the delay existing between the handwriting training and the early signs of alerts that lead to referrals (in France, the diagnostic of dysgraphia is not recommended prior to the second grade).

On the basis of the clinical annotation of the database defining two datasets (TD children vs. children with dysgraphia), a study was performed to investigate children's handwriting acquisition based on BHK clinical assessment [17] and a set of digital features [24].

From the 12 digital features we selected in this work, it was interesting to notice that only three of them were significantly correlated with the BHK handwriting quality score of TD children (without dysgraphia), while all of them were significantly correlated when considering the whole sample (TD and D dataset). In other words, it seems that some of the digital features are not useful in explaining the handwriting quality of TD children. From the three features associated with the handwriting quality of TD children, two of them describe handwriting on a kinematic aspect and one describes such on a static aspect. None of them were features associated with the pressure or tilt aspect of handwriting, while these are strongly associated in children with dysgraphia [24, 29]. These results suggest that the pressure and tilt aspects of handwriting may be particularly central aspects of dysgraphia [24, 26, 27].

In terms of clinical relevance, our findings show the limitation of the current clinical tests used to assess handwriting quality, as the digital features are for the moment neglected due to the technology used (pen and paper tests). They also prove the benefit digital tablets can bring in the assessment of handwriting. In terms of remediation, it may also be particularly interesting to place more emphasis on these aspects of handwriting, for example by using gaming activities designed on digital tablets in which pressure or tilt can be integrated and manipulated. Indeed, in other fields of learning (e.g., emotion recognition; attention), serious games have shown great attraction and clinical interests for children with neurodevelopmental disorders [30, 31]

In contrast with traditional classifications of dysgraphia that are based on children's comorbid problems such as dyslexia, attention deficit or motor-coordination impairment like in the Deuel classification, [23], we established a new clustering of children with dysgraphia based on low-level motor aspects of handwriting including static, kinematic, pressure and tilt features.

The K-means clustering based on our 12 digital features yielded three different subgroups. Cluster 1 gathered children with the less severe cases of dysgraphia, including more girls and those with normal speeds. Clusters 2 and 3 included children with the most severe cases of dysgraphia with a preponderance of boys. Children in cluster 2 presented abnormalities in terms of kinematics and pressure, while children in cluster 3 displayed abnormalities in terms of tilt. We do not know whether the clustering introduced in this paper overlaps with existing classifications and in particular the one proposed by Deuel [23], since the sample size each cluster too small to compare the disorders of the children. We hope to perform such a study in the near future. As expected [15, 17], we found that a majority of children with dysgraphia were boys (71%) [17, 19, 23]. The most severe clusters were also the clusters including more boys. Although being left-handed is believed to be associated with handwriting difficulties in folk psychology [32], we did not find that handedness mediated either BHK scores or clustering of the children with dysgraphia.

The current study needs to be interpreted with consideration of both its strengths and limitations. Digital tablets can measure several features that are relevant to better understand and classify children with dysgraphia. On the basis of the results of this study, we plan to move forward with the development of new handwriting tests capable of running on digital tablets (e.g. an Ipad) for the purpose of helping the diagnosis of handwriting-acquisition deficit. However, the handwriting data used in this study were acquired in an ecological setting since the children wrote on a paper attached on the tablet.

Several studies have shown that the writing sensation (often called friction of the pen) of individuals who write with a pen on a sheet of paper might be very different from the one of individuals who write with a stylus on the surface of a digital tablet [33]. To generalize our results with a stylus/tablet setting, we need first to check whether the same set of features remains relevant. A new database of handwriting traces should be acquired directly on digital tablet with a stylus to investigate whether generalization of our results is possible in this setting.

Secondly, the digital tablets used for the handwriting acquisitions in school and in the specialized clinic were different (Wacom Intuos 3 in specialized clinic vs. Wacom Intuos 4 in school). Despite a careful calibration, the difference in material may have affected some technical aspects of the feature recording. However, to assess whether pressure registration differed between the two tablets, we performed the following experiment: with the pen vertically positioned on the surface of the tablets, 15 different weights were used as an input while the values returned by each tablets were logged. We modelled inputs and outputs in each tablet and between tablets. The correlation found were above 0.99 (p<.001), meaning that the impact was likely limited. For further technical discussion we invite readers to look at the following paper [34].

Finally, it is possible that the transversal design of this study cannot allow for the longitudinal assessment of the development of a typical child or a child with dysgraphia from one grade to another. We believe it will be important to assess the evolution of these digital features longitudinally within the same child during learning or rehabilitation.

## 4 Conclusion

The current results open new opportunities for the automatic detection of children with dysgraphia in classroom. We also believe that the training of pressure and tilt may open new therapeutic opportunities through serious games able to manipulate these features.

## Acknowledgments

We would like to thank Lucile Huet and Elodie Navarre (two juniors psychomotor therapists) for the annotation of the BHK.

## Author Contributions

**Conceptualization:** Thomas Gargot, Pierre Dillenbourg, David Cohen, Caroline Jolly.

**Data curation:** Thomas Gargot, Hugues Pellerin, Ingrid Zammouri, Laurence Casteran, David Cohen, Caroline Jolly.

**Formal analysis:** Thomas Gargot, Hugues Pellerin.

**Methodology:** Thomas Gargot, Ingrid Zammouri.

**Project administration:** Thomas Gargot, David Cohen, Caroline Jolly.

**Resources:** Thibault Asselborn.

**Software:** Thibault Asselborn.

**Supervision:** Thomas Gargot, Ingrid Zammouri, David Cohen.

**Validation:** Thomas Gargot, Thibault Asselborn, Hugues Pellerin, Ingrid Zammouri, David Cohen.

**Visualization:** Thomas Gargot, Thibault Asselborn, Hugues Pellerin.

**Writing – original draft:** Thomas Gargot, Thibault Asselborn, David Cohen.

**Writing – review & editing:** Thomas Gargot, Salvatore M. Anzalone, Wafa Johal, Pierre Dillenbourg, David Cohen, Caroline Jolly.

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
