## [Decision Letter · Decision Letter 0]

12 Feb 2020

PONE-D-19-34910

Acquisition of handwriting in children with and without dysgraphia: a computational approach

PLOS ONE

Dear Dr GARGOT,

Thank you for submitting your manuscript to PLOS ONE. After careful consideration, we feel that it has merit but does not fully meet PLOS ONE’s publication criteria as it currently stands. Therefore, we invite you to submit a revised version of the manuscript that addresses the points raised during the review process.

We would appreciate receiving your revised manuscript by Mar 28 2020 11:59PM. To enhance the reproducibility of your results, we recommend that if applicable you deposit your laboratory protocols in protocols.io, where a protocol can be assigned its own identifier (DOI) such that it can be cited independently in the future. For instructions see: http://journals.plos.org/plosone/s/submission-guidelines#loc-laboratory-protocols

We look forward to receiving your revised manuscript.

Kind regards,

Yih-Kuen Jan, PhD

Academic Editor

PLOS ONE

Journal Requirements:

3. Please amend the manuscript submission data (via Edit Submission) to include author Ingrid Zammouri.

Reviewers' comments:

Reviewer's Responses to Questions

**Comments to the Author**

1. Is the manuscript technically sound, and do the data support the conclusions?

Reviewer #1: Yes

Reviewer #2: Yes

2. Has the statistical analysis been performed appropriately and rigorously? 

Reviewer #1: Yes

Reviewer #2: N/A

3. Have the authors made all data underlying the findings in their manuscript fully available?

Reviewer #1: Yes

Reviewer #2: Yes

4. Is the manuscript presented in an intelligible fashion and written in standard English?

Reviewer #1: Yes

Reviewer #2: No

5. Review Comments to the Author

Reviewer #1: Methods: It was mentioned that there were junior and senior therapists scoring the BHK. However, I did not find any statistical analysis about the agreement or disagreement among those (Kappa?). It would be useful to note this aspect in limitations or explain better this situation in results section.

Results: Delete "only" from page 7, line 209 and page 13 line 377.

Likewise, delete "As expected" from page 7, line 218 since it was not an objective of the study.

Discussion: Delete one of the two "Finally" from page 14 lines 406 and 445.

Reviewer #2: # Overall Comments

1. Quantitative vs qualitative dysgraphia – This paper's usage of "quantitative" and "qualitative" is non-standard, so it should be clarified upfront. In the Methods section, quantitative dysgraphia is defined as "speed of writing" while qualitative dysgraphia is defined as "quality of writing." These should be defined at the beginning of the paper (both in the abstract and in the early introduction). Without this context, it's confusing why static/kinematic/pressure/tilt features (measured quantitatively) are used to describe "qualitative" dysgraphia.

2. Groups – Why were the comparisons/interactions analyzed with TD vs TD+D groups rather than simply TD vs D? In general, this seems like a very indirect way of analyzing the groups. As a specific example (L393-396), it's reported that "None of [the three features associated with the handwriting quality of TD children] were features associated with pressure or tilt aspect of handwriting, while these are strongly associated in children with dysgraphia [24,29]. These results suggest that the pressure and tilt aspects of handwriting may be particularly central aspects of dysgraphia [24,26,27]." It seems like these conclusions about the significance of pressure and tilt in dysgraphia could have been statistically analyzed directly in this study rather than extrapolating from [24,26,27,29].

3. Figure numbering – Most (all?) figure references need to be fixed. The Results and Discussion were difficult to follow because of the mixed-up numbering.

# Line-by-Line Comments

Abstract: "but present a qualitative different development in terms of static, kinematic, pressure, and tilt feature" – These are quantitative features, so the "qualitative" reference is confusing.

L15: "Despite correct training" – Please expand on what is considered "correct" training (or reword). I am not sure about [17] as I cannot find an English version, but [19] used 3 months of physiotherapy. Is that considered objectively correct?

L66: "with a sensibility of 96.6%" – Is this supposed to be sensitivity?

L106: "Pressure data were carefully calibrated between the two tablets" – Please expand on this calibration. Was this a simple linear mapping, or did it require something more complex? Also, consider moving the description of the pressure comparisons (from the Limitations section) to here.

L108: "inter-cotator value of 97%" – Is this referring to inter-rater?

L211: "We considered the children with qualitative dysgraphia as children with dysgraphia in the D dataset the children having qualitative dysgraphia" – Unclear, please clarify

L242: "As expected, since all these features already classified in a proper manner dysgraphia on a binary classification in Asselborn et al. [24]" – Unclear, please clarify

# Copyedits

L9: "handwriting initially evolves first on a qualitative level" – Remove either "first" or "initially" (redundant).

L34/L408/L417: "Deuel et al. [23]" – These should be Deuel [23] (without et al).

L62: "typically developing (TD) and children with dysgraphia" – Reword to "typically developing (TD) children and children with dysgraphia" or "children with typical development (TD) and with dysgraphia."

L128/L142/L156/L166: "They regroup features" – Using "they" is a bit awkward here. Consider "these" or something similar.

L159: "Mean Speed Of Pressure Change" –"Of" should be lowercase.

L192: "the distribution of the residuals were not normally distributed" – "Were" should be "was."

L199: "theoretical classification of Deuel" – In-text citation [23] should be included.

L210: "and all of them showing" – Reword to "with all of them showing" or "and all of them showed."

L219: Extra space before the final period.

Table 4: Extra parenthesis after "quality" and missing parenthesis after "dataset."

6. PLOS authors have the option to publish the peer review history of their article (what does this mean?). If published, this will include your full peer review and any attached files.

Reviewer #1: No

Reviewer #2: No

---

## [Author Response · Author response to Decision Letter 0]

22 Apr 2020

PONE-D-19-34910

Acquisition of handwriting in children with and without dysgraphia: a computational approach

RESPONSE TO REVIEWERS

RESPONSE TO REVIEWER #1:

We thank reviewer 1 for his/her interest and valuable feedbacks regarding our MS. We believe these improved both quality and readability of the revised MS. We addressed all points raised by reviewer 1 as follow.

A formated word document of this response, can be also found at the end of the submission

Point 1. Methods: I did not find any statistical analysis about the agreement.

We thank reviewer 1 for raising this point that was unclear. As scoring was a continuous variable, we used an Intra class Correlation. We clarified this in the revised MS. It is now pages 4-5 of the revised MS: “Two junior psychomotor therapists were trained by the same senior psychomotor therapist to score BHK. Then, the 2 juniors therapists annotated independently all BHK both for quality and speed scores. For the 30 least consistent scores (BHK score > 5), the senior therapist scored the BHK. These professionals were blinded to the demographics and clinical characteristics of the children. Scoring included two dimensions: (1) handwriting velocity assessed through the number of characters written in five minutes and (2) handwriting quality on the five first sentences of the text according to 13 items using a semiquantitative method (BHK quality scores: Table 2). We calculated the final inter rater-reliability using intra-class correlation, ICC = 0.97 (95% CI: 0.96-0.98) Finally, according to the normal scores by age measured during the previous validation of the scale [17], we computed a qualitative score (quality of the writing) and a quantitative score (speed of the writing).”

Point 2. Results: Delete "only" from page 7, line 209 and page 13 line 377.

This was done accordingly in the revised MS

Point 3. Likewise, delete "As expected" from page 7, line 218 since it was not an objective of the study.

This was done accordingly in the revised MS

Point 4. Discussion: Delete one of the two "Finally" from page 14 lines 406 and 445.

These were done accordingly in the revised MS

 

RESPONSE TO REVIEWER #2:

We thank reviewer 2 for his/her interest and valuable feedbacks regarding our MS. We also thank reviewer 2 for his/her careful editing of our MS. We believe these improved both quality and readability of the revised MS. We addressed all points raised by reviewer 1 as follow.

Overall Comments

Point 1. Quantitative vs qualitative dysgraphia – This paper's usage of "quantitative" and "qualitative" is non-standard, so it should be clarified upfront. 

We thank reviewer 2 for this important comment. Indeed, the term qualitative can be confusing. Reviewer 2 is right: all the analyses we performed were quantitative. To clarify that, we replaced when necessary qualitative by quality score which is used for BHK to quantify the quality of writing (as opposed to speed). It is a continuous numerical score described Table 2. We also replaced quantitative by speed. 

Point 2. Groups – Why were the comparisons/interactions analyzed with TD vs TD+D groups rather than simply TD vs D? In general, this seems like a very indirect way of analyzing the groups. 

We thank reviewer 2 for this important comment and we understand his/her concern. A said in the introduction, we selected the 12 digital features through machine learning classifying BHK scores as threshold (binary classification). As machine learning is based on statistical occurrences, we already know that the differences between TD vs. D are significant. To clarify our choice in the revised MS, we made the following revisions:

1. In the introduction after summarizing the first study we performed to select the digital features through machine learning, we added the following paragraph (page 3 of the revised MS): “Despite the inherent progressive learning of writing, so far no study using digital features took into account age and had a developmental approach. We still do not know how the selected features classifying children with dysgraphia evolved in TD children. In addition, we don’t know whether their ability to detect children with dysgraphia changed with age.”

2. We also better distinguish the objectives of the study. It is now page 3 of the revised MS: “In the current study, we aimed to extend our work [24] addressing the effect of age, and the heterogeneity of dysgraphia. Our objectives were the following: 1. First, we aimed to present the learning and acquisition of handwriting from a developmental approach (according to child age). We explored TD children in order to better understand typical development (TD dataset only) and children with dysgraphia (D dataset). 2. Second, we aimed to identify the best features, to diagnose children with dysgraphia (according to age) both using the clinical gold standard method as well as relevant digital features [24]. 3. Third, […]”

3. In the method section we explained this choice in terms of statistics. It is now pages 6-7 of the revised MS: “Since we selected the 12 digital features through machine learning classifying BHK scores as threshold (binary classification), we considered inappropriate to use direct group comparisons (TD vs. D). To take into account the effect of age, and possible interactions between a given digital feature and age, we applied linear regressions considering each feature as a continuous variable to explain the BHK as a continuous variable without consideration of the diagnosis threshold. To do so, each of the 12 digital features was normalized in order to assess their effect in linear regression models. 

To understand how a given digital feature is explaining or not BHK quality taking into account grade and gender, a linear regression model per feature was created to predict the continuous BHK quality score. This model was adjusted for grade and gender. In the same way, a model was created to predict the BHK speed score. The formulas can be described as follows:

To understand how a given digital feature explaining continuous BHK changes according to a child’s grade, a similar model with interaction [grade*Normalized(feature)] was also created. In other words, the model can show the relative importance of a given digital feature to diagnose dysgraphia according to age. As recommended in the BHK manual, we selected the grade rather than the age to assess the effect on education, since the writing process is learned at school and not spontaneously. The formulas can be described as follows:

 Since the distribution of the residuals was not normally distributed, a bootstrap analysis (with 10,000 replications) was performed to assess the 95% confidence intervals (95%CI) and p values. These were respectively obtained by BCA (bias-corrected and accelerated) bootstrap and percentile bootstrap with the R boot package. As said previously, we performed these analyses on the TD dataset only to explore how digital features predict writing (BHK) quality and speed in TD children, then on the TD + D dataset to explore how digital features predict writing (BHK) quality and speed in a mixed population that resembles a more realistic situation in the context of school detection of D children.”

Point 3. Figure numbering – Most (all?) figure references need to be fixed. The Results and Discussion were difficult to follow because of the mixed-up numbering.

We thank reviewer 2 for this. We are sorry for the confusion. All numbering were checked and fixed accordingly

Line-by-Line Comments

Point 1. Abstract: "but present a qualitative different development in terms of static, kinematic, pressure, and tilt feature" – These are quantitative features, so the "qualitative" reference is confusing.

As requested, we replaced qualitative by quality score. It is a continuous numerical BHK score described Table 2. Also, we replaced quantitative by speed. 

Point 2. L15: "Despite correct training" – Please expand on what is considered "correct" training (or reword). I am not sure about [17] as I cannot find an English version, but [19] used 3 months of physiotherapy. Is that considered objectively correct?

Sorry for misusing correct training. We changed in the revised MS for “Despite normal education exposure”

Point 3. L66: "with a sensibility of 96.6%" – Is this supposed to be sensitivity?

As requested, sensibility was changed to sensitivity

Point 4. L106: "Pressure data were carefully calibrated between the two tablets" – Please expand on this calibration. Was this a simple linear mapping, or did it require something more complex? Also, consider moving the description of the pressure comparisons (from the Limitations section) to here

As suggested, a more precise description of the procedure used for the calibration was added to the manuscript. It is now page 4 of the revised MS: “The weights X were carefully chosen (from 0g (pen only) to 400g) in order to explore all range of tablet outputs until saturation. The relation between the weight in input (X) and the value returned by the tablet (Y) could be extracted and was found to be very similar for the two tablets (Spearman correlation > 0.99, p < 0.001, mean square error = 0.6). A 4th degree polynomial fit was created to model the function describing the X/Y relation of the first tablet and used on the second to correct the output. After this correction, the spearman correlation was found to be 0.99998 (p << 0.001) and the mean squared error was 5.1x10-3.”

Point 5. L108: "inter-cotator value of 97%" – Is this referring to inter-rater?

We thank reviewer 2 for this comment. Indeed, we calculated the final inter rater-reliability using intra-class correlation (ICC = 0.97 (95% CI: 0.96-0.98)

Point 6. L211: "We considered the children with qualitative dysgraphia as children with dysgraphia in the D dataset the children having qualitative dysgraphia" – Unclear, please clarify

We thank reviewer 2 for this comment. After clinical assessment of all BHK tests from our dataset (280 children), we confirmed dysgraphia in all children recruited in special clinics and detected 13 (5.63%) children with dysgraphia among those recruited from regular schools. Speed dysgraphia (slow writing) was observed in 12 children with all of them showing also qualitative dysgraphia (poor quality, legibility). Thus, we defined the diagnostic category of dysgraphia based on the BHK quality score only.

This last sentence was added in the revised version of the MS to make clear our point.

Point 7. L242: "As expected, since all these features already classified in a proper manner dysgraphia on a binary classification in Asselborn et al. [24]" – Unclear, please clarify

We thank reviewer 2 for this comment. This point is related to what we said and clarified previously (see point 2 response to overall comments). To clarify this point we changed the sentence in the revised MS as follow (page 5): “In this work, we only used the features that were found to be the most important in the aforementioned random forest model according to the Gini importance metric [24]. This means that all the features were significantly different between TD and D based on a binary diagnostic classification (BHK threshold). As expected, all digital features were significantly associated with continuous BHK quality score when the models were applied on TD children and children with dysgraphia.”

# Copyedits

Point 1. L9: "handwriting initially evolves first on a qualitative level" – Remove either "first" or "initially" (redundant).

This was done accordingly in the revised MS

Point 2. L34/L408/L417: "Deuel et al. [23]" – These should be Deuel [23] (without et al).

This was done accordingly in the revised MS

Point 3. L62: "typically developing (TD) and children with dysgraphia" – Reword to "typically developing (TD) children and children with dysgraphia" or "children with typical development (TD) and with dysgraphia."

This was done accordingly in the revised MS

Point 4. L128/L142/L156/L166: "They regroup features" – Using "they" is a bit awkward here. Consider "these" or something similar.

This was done accordingly in the revised MS

Point 5. L159: "Mean Speed Of Pressure Change" –"Of" should be lowercase.

This was done accordingly in the revised MS

Point 6. L192: "the distribution of the residuals were not normally distributed" – "Were" should be "was."

This was done accordingly in the revised MS

Point 7. L199: "theoretical classification of Deuel" – In-text citation [23] should be included.

This was done accordingly in the revised MS

Point 8. L210: "and all of them showing" – Reword to "with all of them showing" or "and all of them showed."

This was done accordingly in the revised MS

Point 9. L219: Extra space before the final period.

This was done accordingly in the revised MS

Point 10. Table 4: Extra parenthesis after "quality" and missing parenthesis after "dataset."

This was done accordingly in the revised MS

---

## [Decision Letter · Decision Letter 1]

22 Jun 2020

PONE-D-19-34910R1

Acquisition of handwriting in children with and without dysgraphia: a computational approach

PLOS ONE

Dear Dr. GARGOT,

Thank you for submitting your manuscript to PLOS ONE. After careful consideration, we feel that it has merit but does not fully meet PLOS ONE’s publication criteria as it currently stands. Therefore, we invite you to submit a revised version of the manuscript that addresses the points raised during the review process.

We look forward to receiving your revised manuscript.

Kind regards,

Yih-Kuen Jan, PhD

Academic Editor

PLOS ONE

Reviewers' comments:

Reviewer's Responses to Questions

**Comments to the Author**

1. If the authors have adequately addressed your comments raised in a previous round of review and you feel that this manuscript is now acceptable for publication, you may indicate that here to bypass the “Comments to the Author” section, enter your conflict of interest statement in the “Confidential to Editor” section, and submit your "Accept" recommendation.

Reviewer #1: (No Response)

Reviewer #3: All comments have been addressed

2. Is the manuscript technically sound, and do the data support the conclusions?

Reviewer #1: Yes

Reviewer #3: Yes

3. Has the statistical analysis been performed appropriately and rigorously? 

Reviewer #1: Yes

Reviewer #3: Yes

4. Have the authors made all data underlying the findings in their manuscript fully available?

Reviewer #1: No

Reviewer #3: (No Response)

5. Is the manuscript presented in an intelligible fashion and written in standard English?

Reviewer #1: Yes

Reviewer #3: (No Response)

6. Review Comments to the Author

Reviewer #1: Page 2: line 15: delete "normal"

Page 3: line 61: delete "have"

I have problems to understand the number of participants. It was mentioned in Mat & Met "we recruited 280 children." then "Two hundred thirty-one children .." Then, in page 4 says " The 298 children..". Please clarify numbers..

give actual " p values"

In results section (2.1) the first aim does not fit with what was mentioned in page 3. Please reorganize sentences and aims. Likewise, results for aim 1 were mentioned but not for the rest of aims.

Reviewer #3: The discussion about anxiety with dysgraphia needs to explain further.

The use of pen tablets affect vibration on the user is there a description of it? Does this affect the results of the research?

The author uses semi-quantitative, quantitative, and qualitative. This three method makes the reader confused about the research procedures with three ways, and what for the three uses?

Qualitative scoring in this study is not clarified by what method and for what generalizations?

Pen tablet calibration and the tendency of some students for tablet surface vibrations to find a solution in its generalization can be of equal value when calculated.

This research has a good novelty but has weaknesses in the methods and devices used. The exposure to the use of 3 types of ways was not reinforcing in the revised paper. Research on the use of different pen tablets in previous studies does not exist, so this study is a new study using two different devices in series; this needs further discussion.

7. PLOS authors have the option to publish the peer review history of their article (what does this mean?). If published, this will include your full peer review and any attached files.

Reviewer #1: No

Reviewer #3: Yes: Chi-Wen Lung

---

## [Author Response · Author response to Decision Letter 1]

4 Jul 2020

PONE-D-19-34910

Acquisition of handwriting in children with and without dysgraphia: a computational approach

RESPONSE TO REVIEWER #1:

We thank reviewer 1 for his/her interest and valuable feedbacks regarding our MS. We believe that they improved both quality and readability of the revised MS. We addressed all points raised by reviewer 1 as follow.

Reviewer #1: 

Point 1. Page 2: line 15: In the sentence, “Despite normal education exposure, 5% to 10% of children never reach a sufficient 1 level of automation in handwriting” delete "normal". 

As requested we changed the sentence accordingly.

Point 2. Page 3: line 61: delete "have"

As requested we deleted the word ‘have’ in the sentence accordingly.

Point 3. I have problems to understand the number of participants. 

We have detailed the number of participants and group allocation in the Material and Method section and in a diagram flow shown in Figure 2. However, the reviewer is perfectly right there was a mistake in the transcription of numbers when we referred to 298 children. In fact pages 3 and 4 we corrected to the right total number N=280. We thank you for pointing this error.

Point 4. give actual " p values"

Given the complexity of the MS we chose to be conservative with the p values to ease reading (asproposed by PlosOne manuscript recommendation). All p values are reported, but they are shown in all tables 3, 4, 5, 6, 7 reporting the multivariate models and table 8 reporting the cluster analysis. 

Table 1 is a description of the participants and table 2 lists the items used in the BHK test and the automatic features that we extracted. There is no comparison, so no p values need to be reported.

Point 5. CHECK the results in the text: In results section (2.1) the first aim does not fit with what was mentioned in page 3. Please reorganize sentences and aims. Likewise, results for aim 1 were mentioned but not for the rest of aims.

We agree with the reviewer and to help following our aim in the results section, we repeated them accordingly. We added in the revised version of the MS at the beginning of section 2.2 the following sentence: “Our second aim was to identify the best features to diagnose children with dysgraphia both using BHK scores (the clinical gold standard method) and relevant digital features, and to explore how the relevant digital features had statistical interaction with age.” 

In addition, we added at the beginning of section 2.3.: “Our third aim was to explore whether we could define a new classification of dysgraphia based on our automated features. Hence, most of the classifications...”

 

RESPONSE TO REVIEWER #3:

We thank reviewer 3 for his/her interest and valuable feedbacks regarding our MS. We believe that they improved both quality and readability of the revised MS. We addressed all points raised by reviewer 3 as follow.

Overall Comments

Point 1. The discussion about anxiety with dysgraphia needs to explain further.

We thank reviewer 3 for this remark. We added in the introduction two sentences and a reference to develop on the link between dysgraphia and anxiety disorders. It is now page 4 of the revised MS: “Hence, in children with dysgraphia, the low performance in writing induces negative comparisons with others and self-criticizing. As a consequence, it increases school-performance anxiety and can lead to an increased trait anxiety that can persist until adolescence particularly in boys (Sigurdsson, 2002)”.

Sigurdsson, E., Van Os, J., & Fombonne, E. (2002). Are impaired childhood motor skills a risk factor for adolescent anxiety? Results from the 1958 UK birth cohort and the National Child Development Study. American Journal of Psychiatry, 159(6), 1044-1046.

Point 2. The use of pen tablets affect vibration on the user is there a description of it? Does this affect the results of the research?

We thank reviewer 3 for raising this point that was not clear enough obviously. Our manuscript does not study vibration. Some automatic features are related to tremor of the pen extremity but are not related to clinical tremor or vibration. We believe that we induced a confusion when we explain in the discussion section that individuals who write on a paper sheet and individuals who write on the screen of a tablet may have different writing sensation (often call friction of the pen). We changed the related sentences to make it clear. It is now page 15 of the revised MS: “Several studies have shown that the writing sensation (often called friction of the pen) of individuals who write with a pen on a sheet of paper might be very different from the one of individuals who write with a stylus on the surface of a digital tablet [33]. To generalize our results with a stylus/tablet setting, we need first to check whether the same set of features remains relevant. A new database of handwriting traces should be acquired directly on digital tablet with a stylus to investigate whether generalization of our results is possible in this setting.”

Point 3. The author uses semi-quantitative, quantitative, and qualitative. This three method makes the reader confused about the research procedures with three ways, and what for the three uses? Qualitative scoring in this study is not clarified by what method and for what generalizations?

We thank again reviewer 3 for raising this point. Hence, we only used quantitative measures (meaning scores) in this study. Otherwise we could not perform linear regressions. The confusion comes from the BHK test that produces 2 scores: one for handwriting speed and the other for handwriting quality. To avoid further confusion, we systematically revised the MS to specify handwriting quality score when needed instead of quality alone as despite a first round of revision we left some occurrences, and we are sorry for that. The changes appear in the ‘revised version modification in yellow’ document.

Point 4. Pen tablet calibration and the tendency of some students for tablet surface vibrations to find a solution in its generalization can be of equal value when calculated.

Again, we believe we introduced a confusing argument in the discussion. See previous response to point 2.

Point 5. This research has a good novelty but has weaknesses in the methods and devices used. The exposure to the use of 3 types of ways was not reinforcing in the revised paper. 

We thank reviewer 3 for this positive remark. Again when one use automated features it is important to have them anchored in clinical practice. When it comes to handwriting as we explained it in the introduction, the BHK test is the clinical gold standard and produces 2 score of equal importance that are the quality and the speed of handwriting. Pupils at school are expected to master both.

Point 6. Research on the use of different pen tablets in previous studies does not exist, so this study is a new study using two different devices in series; this needs further discussion.

We thank reviewer 3 for this important remark. We gave in the discussion the main measures and controls we performed to show the minimal impact of using two different models of the same Wacom tablet. For further technical discussion we invite readers to look at the following paper: Asselborn T, Gargot T, Kidzinski Ł, Johal W, Cohen D, Jolly C, et al. Reply: Limitations in the creation of an automatic diagnosis tool for dysgraphia. npj Digital Medicine. 2019;2(1):37.

---

## [Decision Letter · Decision Letter 2]

30 Jul 2020

Acquisition of handwriting in children with and without dysgraphia: a computational approach

PONE-D-19-34910R2

Dear Dr. GARGOT,

We’re pleased to inform you that your manuscript has been judged scientifically suitable for publication and will be formally accepted for publication once it meets all outstanding technical requirements.

Kind regards,

Yih-Kuen Jan, PhD

Academic Editor

PLOS ONE

Additional Editor Comments (optional):

Reviewers' comments:

Reviewer's Responses to Questions

**Comments to the Author**

1. If the authors have adequately addressed your comments raised in a previous round of review and you feel that this manuscript is now acceptable for publication, you may indicate that here to bypass the “Comments to the Author” section, enter your conflict of interest statement in the “Confidential to Editor” section, and submit your "Accept" recommendation.

Reviewer #1: All comments have been addressed

Reviewer #3: All comments have been addressed

2. Is the manuscript technically sound, and do the data support the conclusions?

Reviewer #1: Yes

Reviewer #3: Yes

3. Has the statistical analysis been performed appropriately and rigorously? 

Reviewer #1: Yes

Reviewer #3: Yes

4. Have the authors made all data underlying the findings in their manuscript fully available?

Reviewer #1: Yes

Reviewer #3: Yes

5. Is the manuscript presented in an intelligible fashion and written in standard English?

Reviewer #1: Yes

Reviewer #3: Yes

6. Review Comments to the Author

Reviewer #1: all queries were addressed by he authors. I recommend the article for publication. I have not more coments.

Reviewer #3: All my comments have been addressed in the revised manuscript, and I recommend its publication. The authors may consider refining the account of the table (8 tables) and figure (6 figures).

7. PLOS authors have the option to publish the peer review history of their article (what does this mean?). If published, this will include your full peer review and any attached files.

Reviewer #1: **Yes: **Fidias E Leon-Sarmiento

Reviewer #3: **Yes: **Chi-Wen Lung

---

## [Editor Report · Acceptance letter]

7 Aug 2020

PONE-D-19-34910R2 

Acquisition of handwriting in children with and without dysgraphia: a computational approach 

Dear Dr. GARGOT:

I'm pleased to inform you that your manuscript has been deemed suitable for publication in PLOS ONE. Congratulations! Your manuscript is now with our production department. 

Kind regards, 

on behalf of

Dr. Yih-Kuen Jan 

Academic Editor

PLOS ONE